

# **Improving Statistical Projections of Ocean Dynamic Sea-level Change**
# **Using Pattern Recognition Techniques.**
Víctor Malagón-Santos[1], Aimée B.A. Slangen[1], Tim H.J. Hermans[1,2], Sönke Dangendorf[3], Marta
Marcos[4], Nicola Maher[5,6,7].
[1]NIOZ Royal Netherlands Institute for Sea Research, Department of Estuarine & Delta Systems, P.O. Box 140, 4400 AC
Yerseke, the Netherlands.
[2]University of Utrecht, Institute for Marine and Atmospheric research Utrecht (IMAU), Utrecht, The Netherlands
[3]Department of River-Coastal Science and Engineering, Tulane University, New Orleans, USA.
[4]Mediterranean Institute for Advanced Studies (IMEDEA), Spanish National Research Council-University of Balearic Islands
(CSIC-UIB), Esporles, Spain.
[5]Cooperative Institute for Research in Environmental Science, University of Colorado, Boulder, CO, USA.
[6]Department of Atmospheric and Oceanic Sciences, University of Colorado, Boulder, CO, USA.
[7]Max Planck Institute for Meteorology, Hamburg, Germany.

*Correspondence to*: Víctor Malagón-Santos (victor.malagon.santos@nioz.nl)
**Abstract.** Regional emulation tools based on statistical relationships, such as pattern scaling, provide a computationally
inexpensive way of projecting ocean dynamic sea-level change for a broad range of climate change scenarios. Such approaches
usually require a careful selection of one or more predictor variables of climate change so that the statistical model is properly
optimized. Even when appropriate predictors have been selected, spatiotemporal oscillations driven by internal climate
variability can be a large source of model disagreement. Using pattern recognition techniques that exploit spatial covariance
information can effectively reduce internal variability in simulations of ocean dynamic sea level, significantly reducing random
errors in regional emulation tools. Here, we test two pattern recognition methods based on Empirical Orthogonal Functions
(EOF), namely signal-to-noise maximising EOF pattern filtering and low-frequency component analysis, for their ability to
reduce errors in pattern scaling of ocean dynamic sea-level change. These two methods are applied to the initial-condition
large ensemble MPI-GE, so that internal variability is optimally characterized while avoiding model biases. We show that
pattern filtering provides an efficient way of reducing errors compared to other conventional approaches such as a simple
ensemble average. For instance, filtering only two realizations by characterising their common response to external forcing
reduces the random error by almost 60%, a reduction level that is only achieved by averaging at least 12 realizations. We
further investigate the applicability of both methods to single realization modelling experiments, including four CMIP5
simulations for comparison with previous regional emulation analyses. Pattern scaling leads to a varying degree of error
reduction depending on the model and scenario, ranging from more than 20% to about 70% reduction in global-mean root-



mean-squared error compared with unfiltered simulations. Our results highlight the relevance of pattern recognition methods
as a tool to reduce errors in regional emulation tools of ocean dynamic sea-level change, especially when one or a few
realizations are available. Removing internal variability prior to tuning regional emulation tools can optimize the performance
of the statistical model and simplify the choice of suitable predictors.
**1    Introduction**
Sea levels are closely linked to the state of the climate. Understanding how increased radiative forcing in the atmosphere will
affect sea-level rise is of utmost importance given the devastating impacts to coastal systems. Global-mean sea level has been
increasing over the 20$^{th}$ century (Fox-Kemper, 2021), and its rate has been accelerating over the past decades both globally
(e.g., Dangendorf et al., 2019; Fox-Kemper, 2021; Frederikse et al., 2020; Nerem et al., 2006) and regionally (e.g., Steffelbauer
et al., 2022). This acceleration is expected to continue over the next century for all greenhouse gas (GHG) concentration
scenarios (Fox-Kemper et al., 2021) with the potential to further increase widespread impacts in coastal areas (Cooley et al.,
2022). Increased sea levels will change coastal flood risk through expanding areas under permanent inundation, increasing
frequencies of extreme coastal flooding events (Vitousek et al., 2017; Wahl et al., 2017), and modifying tides (Haigh et al.,
2020) and thus potentially increasing the frequency of tidal-induced flooding (Moftakhari et al., 2015). These processes will
not only impact coastal infrastructure and assets (Hinkel et al., 2014) but also alter coastal ecosystems and the services they
provide, from ecosystem value to natural flood risk protection (Cooley et al., 2022). Understanding how global and regional
sea levels evolve under different scenarios will help to better adapt to changing risks and mitigate their potential impacts in
coastal zones (Haasnoot et al., 2019, 2021).
Global-mean sea-level change is driven by a combination of processes. The melting of Greenland's and Antarctica's ice sheets,
glaciers and ice caps, changes in land-water storage, and thermal expansion of the ocean are the processes driving global mean
sea-level rise (e.g., Gregory et al., 2019; Fox-Kemper, 2021). Analogously to global warming, sea-level rise is a global concern
but it is not spatially uniform (e.g., Slangen et al., 2017). Four main processes exist that determine regional sea-level change.
First, the redistribution of mass on the Earth's surface, as a result of melting land ice and changes in land-water storage, causes
a regionally variable sea-level change due to gravitational, rotational, and deformational effects (Farrell and Clark, 1976;
Mitrovica et al., 2001). Second, vertical land motion also controls unequal changes in relative sea levels. The viscoelastic
relaxation of the Earth induced by deglaciation following the last glacial maximum, defined as glacial isostatic adjustment
(e.g., Peltier, 1999, 2001) and more local processes driving subsidence (e.g., Nicholls et al., 2021), are the main processes
driving changes in land elevation. Third, (partly wind-driven) ocean circulation, and heat and freshwater fluxes over the ocean,
also known as ocean dynamics (Gregory et al., 2019), change local densities and move water mass around the ocean. Fourth,
changes in sea-level pressure over the oceans, also known as inverted barometer (IB) effects, may lead to regionally varying
rates of sea-level change (Stammer and Hüttemann, 2008)



This study focuses on ocean dynamic sea-level (DSL) change, which is governed by changes in ocean circulation and density.
DSL is strongly influenced by natural variability, and typically contains the largest spatiotemporal variability among all the
regional sea-level change components. These characteristics make it a crucial component to predict regional sea-level changes
accurately, yet also one that provides significant uncertainty (Couldrey et al., 2021). The effect of climate change on DSL is
typically simulated with General Circulation Models (GCMs), which solve a range of geophysical variables controlling the
Earth's climate system. However, GCMs require vast computational resources, and therefore climate modelling experiments
have been designed for a limited range of GHG concentration scenarios (O'Neill et al., 2017; Riahi et al., 2017; van Vuuren
et al., 2011) within the climate model intercomparison (CMIP) framework (Eyring et al., 2016), so that model differences are
somewhat comparable.
To reduce the computational demand, several complementary approaches based on statistical modelling have been proposed.
For instance, regional emulation tools provide a computationally inexpensive alternative for projecting a regional variable and
assessing its response to different forcings. One of the most commonly used emulation approaches for projecting changes in a
regional variable is pattern scaling (Mitchell, 2003; Perrette et al., 2013; Santer et al., 1990), which consists of relating a local,
grid-point variable (predictand) to one or a few global-mean change variables (predictors) via regression. Based on that
statistical relationship, a change in a regional variable can be emulated by projecting the global-mean variables via simpler
climate models (Goodwin et al., 2018; Meinshausen et al., 2011; Millar et al., 2017; Smith et al., 2018)
Here, we build on the approach proposed by Bilbao et al. (2015), who applied a linear pattern scaling approach to assess the
ensemble mean DSL computed from five CMIP5 models and their simulations of several variables describing global changes,
including Global Surface Air Temperature (GSAT), Global-Mean Thermosteric Sea-Level Rise (GMTSLR), and ocean-
volume mean temperature. While GSAT turned out to be the best predictor of 21$^{st}$-century DSL change in a high emissions
scenario (Representative Concentration Pathway (RCP) 8.5), ocean-volume mean temperature and GMTSLR outperformed
the rest of variables considered in lower emissions scenarios (RCP 2.6 and 4.5). As the surface ocean layer responds quicker
to air temperature changes than the deeper ocean layer, they speculated that surface warming had a more important role relative
to deep warming in a high emissions scenario. Based on Bilbao et al. (2015)'s findings, Yuan and Kopp (2021) used the same
set of CMIP5 models to develop a bivariate pattern scaling approach, accounting for the surface and deep ocean layers
separately. Their goal was to capture the different delayed response of those two layers by using GSAT and global-mean deep
ocean temperature changes as predictors. By employing a bivariate pattern scaling approach, Yuan and Kopp (2021) reported
a reduction of the predicted DSL error for the period 2271-2290 of 36%, 24%, and 34% for RCP 2.6, 4.5, and 8.5, respectively,
compared to a univariate approach based on only GSAT.
The aforementioned studies highlight the importance of selecting appropriate predictors to attain an optimized regional
emulator of DSL, and how accounting for different processes driving DSL change (in different layers of the ocean) can help
further improve emulator performance. While designing a regional emulator based on performance metrics may provide
insights into the global processes driving DSL changes, this process can be obscured by other drivers of emulator error. In
particular, random errors contained in the regression forming the pattern scaling approach, are assumed to be mostly caused





by internal climate variability (Bilbao et al., 2015) and may be a source of large uncertainty. Thus, if random errors are not
minimized prior to emulator training with GCM simulations, their presence could impair a proper selection of global predictors,
such that it would be uncertain whether an increase in model performance is due to an appropriate selection of predictors or
an artifact of natural variability causing a biased selection. In previous studies, this effect has been minimized by computing
30-year means, assuming this cancels out natural variability. This step, however, entails a substantial loss of data and does not
guarantee natural variability is optimally subtracted, and residual natural variability, for instance caused by long-memory
processes (e.g., Becker et al., 2014; Dangendorf et al., 2014), can remain.
We therefore propose to take a different approach to separate internal variability from the response driven by external radiative
forcing in the Earth, by employing state-of-the-art modelling experiments specifically designed to do so. These are known as
Single-Model Initial Condition Large Ensembles (SMILES) and consist of a set of simulations with the same forcing but with
the variability evolving in a different phase (Deser et al., 2020). These realizations can be combined through different methods
(e.g., Frankcombe et al., 2015) so that internal variability cancels out. However, conventional approaches such as computing
the ensemble mean or linear trends are not the most efficient tools to do so and tend to lead to the loss of much of the
information gained from running large ensembles (Wills et al., 2020). Other methods based on pattern recognition via
Empirical Orthogonal Functions (EOFs) exploit spatial covariance information to remove internal variability more efficiently
(Wills et al., 2020) and have demonstrated to provide a superior agreement between observations and simulations than an
ensemble average (Marcos and Amores, 2014). These types of efficient methods for removing internal variability hold potential
to benefit emulation experiments of DSL for which the number of simulations is limited.
The aim of this study is to characterise the importance of natural variability as a driver of random errors in statistically based
(pattern-scaled) projections of DSL change. To achieve this aim, we will compare different pattern recognition techniques,
including Signal-to-Noise Maximising (S/N M) EOF pattern filtering (Wills et al., 2020) and Low Frequency Component
Analysis (LFCA, Wills et al., 2018, 2020). We will use these techniques to truncate natural variability in DSL simulations
from the MPI-GE SMILE (Maher et al., 2019), and explore their applicability to single realization modelling experiments,
including a set of CMIP5 simulations used in previous pattern scaling studies. In this paper, we particularly aim to attain the
following objectives:

1)  Use a large ensemble (MPI-GE) to determine the forced pattern and examine to which extent pattern recognition

126          techniques isolate the forced response in DSL change more efficiently than conventional methods (Section 4.1)

2)  Determine the error reduction in pattern scaling of DSL provided by pattern recognition methods relative to more

128          conventional methods (Section 4.2).

3)  Test whether filtering improves pattern scaling in single-realization modelling experiments of DSL (Section 4.3).



## 2 Climate model data and pre-processing

Separating natural variability from the forced response is key for detection and attribution studies in climate change (Labe and Barnes, 2021) and to understand its effects on the climate system (Deser et al., 2020; Mankin et al., 2020). However, the combination of distinct GCMs to analyse internal variability should be performed with caution, as this may conflate internal variability with model biases (Maher et al., 2021b). In recent literature, this has motivated the development and use of SMILES, which branch each realization at a different model stage in the pre-industrial control simulation (Danabasoglu et al., 2020; Deser et al., 2020; Fasullo et al., 2020; Kay et al., 2015; Maher et al., 2019, 2021a; Mankin et al., 2020). This results in simulations with the same forced response but with variability evolving in a different phase, enabling a separation of the variability from the forced response.

From the available SMILES (Deser et al., 2020; Maher et al., 2021a), we decided to use the Max-Planck Institute Grand Ensemble (MPI-GE; Maher et al., 2019) because it contains the largest number of ensemble members available (100) in a SMILE. Moreover, the realizations are available for historical simulations and different RCP scenarios (RCP 2.6, 4.5, and 8.5) up to 2100 together with an extended pre-industrial control simulation. The MPI-GE ensemble design is based on macro-initialization, where 100 distinct coupled initial conditions are sampled from well separated starting dates in the pre-industrial control, such that ensemble members start from different ocean and atmospheric states. This procedure allows assessing uncertainty due to initial conditions differences in large scale aspects of the climate system as well as uncertainty in future model climate due to the non-linear nature of the climate system (Hawkins et al., 2016; Stainforth et al., 2007). Macro-initialized ensembles are therefore better suited than 'micro' ensembles, which are the ones where atmospheric initial conditions are perturbed, to sample uncertainty in an initialized framework, facilitating an assessment of natural variability within a model.

Additionally, we use four CMIP5 models that were used in previous studies of DSL pattern scaling (Bilbao et al., 2015; Yuan and Kopp, 2021), including GISS-E2-R, HadGEM2-ES, IPSL-CM5A-LR, and MPI-ESM-LR. These four GCMs were selected in the afore-mentioned studies because they were used to calibrate the parameters of the simple climate model used by Geoffroy et al. (2013a, b), which facilitated the design of their emulation tool. Also, these models provide multi-century data (up to 2300) in three emissions scenarios, granting an assessment of the suitability of pattern scaling for long-term projections. We use them here for comparison purposes.

The focus of this study is on DSL, which in CMIP models is also known as 'zos' (Griffies et al., 2016) and defined at each location and time as the difference between local sea-surface height relative to the geoid, and its global mean over the ocean area (GMTSLR, or 'zostoga' in CMIP experiments). Hence, by definition, DSL, or zos, varies locally due to ocean circulation and horizontal gradients, but its global mean is zero at every time step. Both zos and zostoga are often expressed in terms of changes relative to a control state, expressing them as differences in relation to a baseline period. Moreover, sea level is influenced by atmospheric pressure anomalies, which is known as the IB effect. DSL simulations from GCMs do not include



the effect of sea-level pressure on sea level and such effect is not subject of study in our analysis, hence it is not considered
here.
Since we are interested in assessing the forced response in DSL for historical and future GHG emissions we will use zos from
a range of GCMs for historical and future radiative forcing scenarios, including RCP 2.6, 4.5, and 8.5 (Meinshausen et al.,
2011). Once the forced DSL has been characterized, we will proceed to pattern scale each model and scenario using GMTSLR
(zostoga) from their respective GCM simulation. Among other potential global predictors, we chose GMTSLR as it is closely
related to DSL, and it has been successfully used in previous pattern scaling analysis of DSL (e.g., Bilbao et al., 2015; Thomas
and Lin, 2018). We refrain from testing other global variables as predictors to ease comparing models and scenarios, and
determining to which extent pattern filtering reduces statistical error via reducing internal variability.
In this study, we are particularly interested in removing interannual variability, thus we compute annual mean zostoga and zos
time series from the raw monthly mean GCM data. In addition, since GCMs are run for a few centuries and the deep ocean
usually takes millennia to reach an equilibrium, both zos and zostoga are subject to model drift (Sen Gupta et al., 2013). Model
drift in the historical and scenario simulations can be corrected for by subtracting the smoothed long-term change of the pre-
industrial control run. To avoid contaminating the drift correction with natural variability, ideally the full length of the control
run is used to determine the drift (Sen Gupta et al., 2013). Therefore, to dedrift the historical and scenario simulations of
zostoga and zos (the latter on a grid cell by grid cell basis) we first fit a quadratic polynomial to the full pre-industrial control
simulations of these variables. Then, we evaluate and subtract the polynomial fit over the time period in which the pre-industrial
control run and historical and scenario runs overlap, as identified by the branch times of the different simulation realizations
and their length, from the historical and scenario runs. Similar to what was found by Hermans et al., (2020) and Hobbs et al.
(2016), fitting a linear or quadratic polynomial to the pre-industrial control simulations yields little difference for the drift-
correction of the zostoga simulations of GISS-E2-R, HadGEM2-ES, IPSL-CM5A-LR, and MPI-ESM-LR. However, in the
pre-industrial simulation of MPI-GE, the increase of zostoga behaves non-linearly and levels off toward the branching time of
ensemble member 40, so we only dedrift ensemble members 1 to 39. For zos, some differences are found between linear and
quadratic drift correction depending on the model, variant, and location. We assume linear dedrifting is suitable for our
purpose, since we verified that the dedrifting does not substantially affect the pattern scaling performance and it is tedious to
assess the best fit on a grid-point basis. After dedrifting, the area-weighted mean of zos is removed at each timestep, and the
resulting fields are bilinearly regridded to a common 1 by 1 degree grid.

## 3 Methods

### 3.1 Pattern recognition techniques

Both S/N M EOF pattern filtering and LFCA are based on a technique called linear discriminant analysis to identify spatial
features, as defined by linear combinations of EOFs, that maximize a type of variance describing a 'signal'. This technique
allows to distinguish that signal from noise existing due to internal variability or between realizations. The difference between




S/N M EOF and LFCA lies in their definition of what type of variance constitutes the signal and the noise. The following subsections briefly explain the pattern filtering methods mentioned above. For a more detailed description, the reader is referred to Wills et al. (2018, 2020).

### 3.1.1 S/N-maximizing pattern filtering

S/N M EOF pattern filtering detects anomaly patterns that maximize a particular type of variance defined as signal (Schneider and Griffies, 1999; Ting et al., 2009; Venzke et al., 1999). One way of doing so is to assess a simulation of forced climate change relative to a preindustrial control (DelSole et al., 2011; Marcos and Amores, 2014). This is advantageous in single-realization GCM experiments, as it only requires one forced realization and one preindustrial control run. However, this could neglect the forced response when external forcing only affects the phase of a mode of internal variability (Wills et al., 2020). Another way of applying this method, which would avoid phase neglection issues, is to use various SMILE realizations with the same forcing to find anomaly patterns where different ensemble members agree on the temporal evolution. These patterns with high signal-to-noise ratio form the forced response as simulated by the GCM, and the variability not described by those can be then truncated from the dataset. S/N M EOF pattern filtering finds anomaly patterns associated with time series $t_k$ that maximize the ratio of (ensemble mean) signal to total variance:

$$s_k = \frac{\langle t_k \rangle^T \langle t_k \rangle}{t_k^T t_k},$$  (1)

where angle brackets represent an ensemble average and k represents a given pattern. These time series (l.h.s. of Fig. 2, for example) are determined by projecting a fingerprint pattern $u_k$ obtained from the EOF analysis (see below) (r.h.s. of Fig. 2, for example) onto the ensemble data matrix $X$:

$$t_k = X u_k,$$  (2)

The ensemble data matrix $X$ is defined as n · $n_e$ × p, where n is the length of time series, $n_e$ is the number of ensemble members, and p is the spatial dimension (i.e., longitude and latitude), and is created by concatenating the $n_e$ × p data matrices $X_i$ from each ensemble member (i) in the time dimension. All data matrices $X_i$ from each ensemble member are weighted by the square root of grid cell area, so that the covariance matrix is also area weighted.

The identified patterns should correspond to variability that occurs within the ensemble. To abide by this requirement, the fingerprint patterns $u_k$ are defined as linear combinations of the N leading ensemble EOFs $a_k$, with normalized weight vectors $e_k$, being $\sigma$ the standard deviation (SD):



$$u_k = \left[\frac{a_1}{\sigma_1} \frac{a_2}{\sigma_2} \cdots \frac{a_N}{\sigma_N}\right] e_k, \qquad (3)$$

Linear-combination coefficients $e_k$ that give $u_k$ and $t_k$ that maximize $s_k$ can be obtained by combining (2) and (3) into (1) (see
Wills et al. (2020) for more details on solving the ensemble EOFs $a_1$ eigenvectors of ensemble mean covariance). The S/N
maximizing patterns $v_k$ are estimated by regressing the ensemble data matrix $X$ onto each $t_k$:

$$v_k = X^T t_k = X^T X u_k = [\sigma_1 a_1 \ \sigma_2 a_2 \ \cdots \ \sigma_N a_N] e_k, \qquad (4)$$

The S/N patterns (S/N Ps) are sorted by the ratio $s_k$, obtaining the leading S/NPs patterns of forced response within the
ensemble. Once the leading S/NPs have been determined, a dataset that excludes internal variability and contains the forced
response ($X_{S/NP}$) can be constructed by truncating the patterns and associated temporal response from the M leading S/NPs:

$$X_{S/NP} = \sum_{k=1}^{M} t_k v_k^T, \qquad (5)$$

To apply S/N M EOF pattern filtering, we must choose two parameters: 1) the number of EOFs retained (N), and 2) the number
of S/N Ps used to compose the forced response (M). Following the approach by Wills et al. (2020), we choose N to retain
between 75% and 95% of the total variance. We use a block bootstrapping approach to determine M, which consists of taking
block (for example, decadal) samples with replacement from the ensemble members to construct dimensionally equivalent
randomized ensembles where the forced response timing of their realizations should not agree with one another. Here, we
choose 30-yr blocks to distinguish forced patterns from internal variability, so that most of internal variability in DSL is
excluded. S/N EOF pattern filtering is then applied to randomized ensembles and the $s_k$ value of the pattern with the highest
S/N ratio is taken as a threshold. This allows us to obtain a distribution of $s_k$ values (one for each randomized ensemble
produced) from which a desired confidence level can be estimated. S/N M EOF patterns with a higher $s_k$ value than the
threshold can be considered as part of the forced response with the chosen confidence level. In contrast, there is no sufficient
statistical evidence to include patterns with a lower $s_k$ value in the forced response, and those are considered noise (internal
variability).
**3.1.2 Low Frequency Component Analysis**
While S/N M EOF has demonstrated to be particularly useful to isolating the forced response when various ensemble members
are available, the advantage of LFCA is that it can analyse the forced response in a single ensemble member (Schneider and
Held, 2001; Wills et al., 2018). The failure to detect some signals in surface air temperature (Wills et al., 2020), such as those
driven by volcanic activity and in some changes in the seasonal cycle is the main disadvantage of this method being document
in literature.



LFCA uses a similar methodology as S/NP M EOF pattern filtering, but detects anomaly patterns associated with time series
$t_k$ that maximize the ratio of low-frequency signal to total variance:

$r_k = \frac{\widetilde{t_k}^T \widetilde{t_k}}{t_k^T t_k},$  ( 6 )

Variations that make it through a low-pass filter (denoted by a tilde), constitute the low-frequency signal (forced response).
Here, we apply a linear Lanczos filter (Duchon, 1979) with a 30-yr lowpass filter, so only longer-scale variability is included.
Similar to S/N M EOF, patterns $v_k$ and their time series $t_k$ are determined by Eqs. (4) and (2), respectively. Low frequency
patterns are then sorted by the ratio $r_k$, so that the leading patterns are those that maximize the ratio of low-frequency to total
variance. The leading anomaly patterns are used to construct a dataset $X_{LFP}$ containing only the variability captured by the
leading M low frequency patterns:

$X_{LFP} = \sum_{k=1}^{M} t_k v_k^T,$  ( 7 )

**3.2 Pattern scaling**
Pattern scaling is usually based on grid-point regression against a global variable, and it assumes that a regional change in DSL
can be explained by global changes of the predictor(s) of choice. Previous studies have shown such relationships can be a
reasonable approximation for different variables of the climate system. For instance, local surface air temperature change
(Collins et al., 2013; Hawkins and Sutton, 2012) and local precipitation (Osborn et al., 2016) have successfully been linked to
GSAT change. Regional emulation based on pattern scaling assumes that patterns of local response to external forcing remains
constant (Tebaldi and Arblaster, 2014), an assumption that can lead to errors (Wells et al., 2022). However, its simplicity and
transferability to many regional variables have made it a popular approach for exploring regional changes in climate change
studies (Bilbao et al., 2015; Fox-Kemper, 2021; Herger et al., 2015; Mitchell, 2003; Osborn et al., 2016; Perrette et al., 2013;
Tebaldi and Arblaster, 2014; Thomas and Lin, 2018; Wells et al., 2022; Wu et al., 2021; Yuan and Kopp, 2021).
Once we have identified the forced DSL within an ensemble of realizations or a single simulation (as outlined in Section 3.1),
we will use this forced response as a predictand in our statistical model for projecting regional DSL. There are different forms
of pattern scaling, mostly differing in the number of predictors included in the analysis (e.g., univariate, Bilbao et al., 2015;
bivariate, Yuan & Kopp, 2021). Here, for simplicity and to ease comparison between raw (de-drifted) DSL and its pattern-
filtered equivalent, we only test pattern scaling based on GMTSLR (or zostoga) as a predictor. The univariate case of pattern
scaling for relating DSL with GMTSLR can be described by the following linear regression relationship:



$$\zeta(t, x, y) = \alpha(x, y)\,\bar{\eta}(t) + b(x, y) + \varepsilon(t, x, y)$$ ( 10 )

Where $\zeta$ and $\bar{\eta}$ denote DSL and GMTSLR, respectively. Longitude and latitude are represented by x and y, whereas t denotes
time. $\alpha$ is a spatial pattern that captures the scaling relationship between DSL and GMTSLR, and b is an intercept term, both
being only a function of location. $\varepsilon$ is a residual term regarded as random noise and often assumed to be driven by internally
generated variability (Bilbao et al, 2015).
**4 Results & Discussion**
**4.1 Forced response in MPI-GE and efficiency of pattern filtering.**
In this section, we focus on determining the forced response in DSL within a SMILE (MPI-GE) using S/N M EOF pattern
filtering and show the efficiency of the latter to remove internal variability compared to conventional approaches. To construct
the forced response based on S/N P, we follow the block-bootstrapping approach described in Section 3.1.1. we define blocks
in terms of thirty years, so most natural variability in DSL is excluded. 30-yr block samples are taken from the 100 historical
realizations of the MPI-GE to construct 20 randomized ensembles. A value of 20 is chosen because increasing it further do not
lead to substantial changes in the estimation of the 95$^{th}$ percentile of $S_k$. The estimated ratio $S_k$ (Eq. 1) for a 95 % confidence
level is 0.08, leading to a total of eight patterns that can be considered as part of the forced response at such a confidence level
(Figure 1).

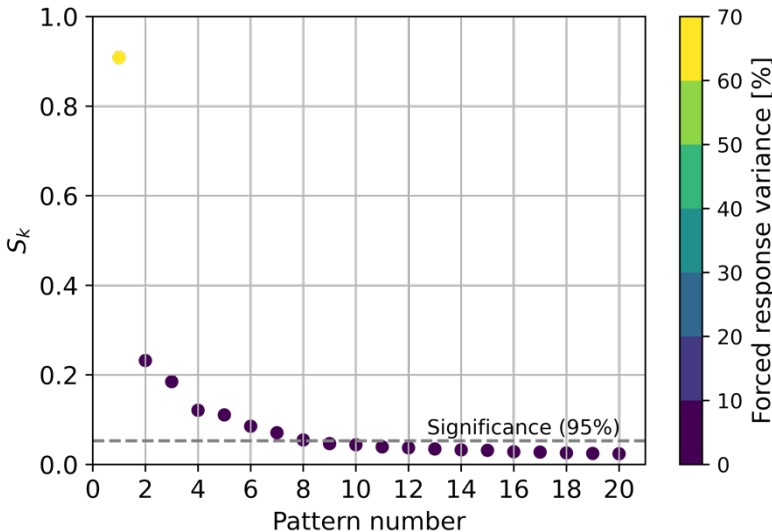





**Figure 1**: Signal fraction of the leading S/N M EOF patterns along with their respective explained forced response variance. The significance level (95%) computed using 30-year block-bootstrapping is represented as a dashed line.

Even though patterns constructed based on EOFs are created from mathematical constraints, known physical processes can be identified in some patterns. For instance, the S/N M EOF pattern with the highest $S_k$ value pattern 1, Fig. 2) explains 62% of the forced response variance (Fig. 1) and is similar to the main forced pattern of DSL change field driven by increased radiative forcing due to increased GHG emissions. There is a zonal dipole in the Southern Ocean, with decreased and increased sea level relative to the mean below and above 50°S, respectively (e.g., Frankcombe et al., 2013). Another dipole structure is found in the North Atlantic with a decreased DSL in the north compared to an increased DSL in the southern section, a feature which appears to disagree with some models (e.g., Bouttes et al., 2014). Nonetheless, the North Atlantic Ocean is an area of large model spread in both CMIP5 and CMIP6 models (Lyu et al., 2020), which suggests the representation of such zonal dipole may be model dependent. Other relevant features include a large DSL rise in the Beaufort Sea and an increased DSL in the North-West Pacific Ocean. Most of these features agree with those documented among CMIP6 and earlier models (Church et al., 2013; Ferrero et al., 2021; Landerer et al., 2007; Lowe and Gregory, 2006; Lyu et al., 2020; Slangen et al., 2014). Patterns are similar between RCP scenarios, mainly differing on their intensity.

The three following resulting patterns (patterns 2, 3 and 4, Fig. S1, S2 and S3) represent between 4-1% (Fig. 1) of the forced response variance and, although with a much lower importance than pattern 1, when combined together represent non-linear processes that start to have an effect in DSL after 2050. Patterns 5, 6, 7 and 8 (Fig. S4, S5, S6, and S7) explain between 1-0.7% of the forced response variance (Fig. 1) and show small perturbations and regional responses that appear to be linked to volcanic eruptions. The patterns number 9 and beyond explain a variance of less than 0,6% and, since their $S_k$ value is not statistically significant at the 95% level, they could be caused by random chance.





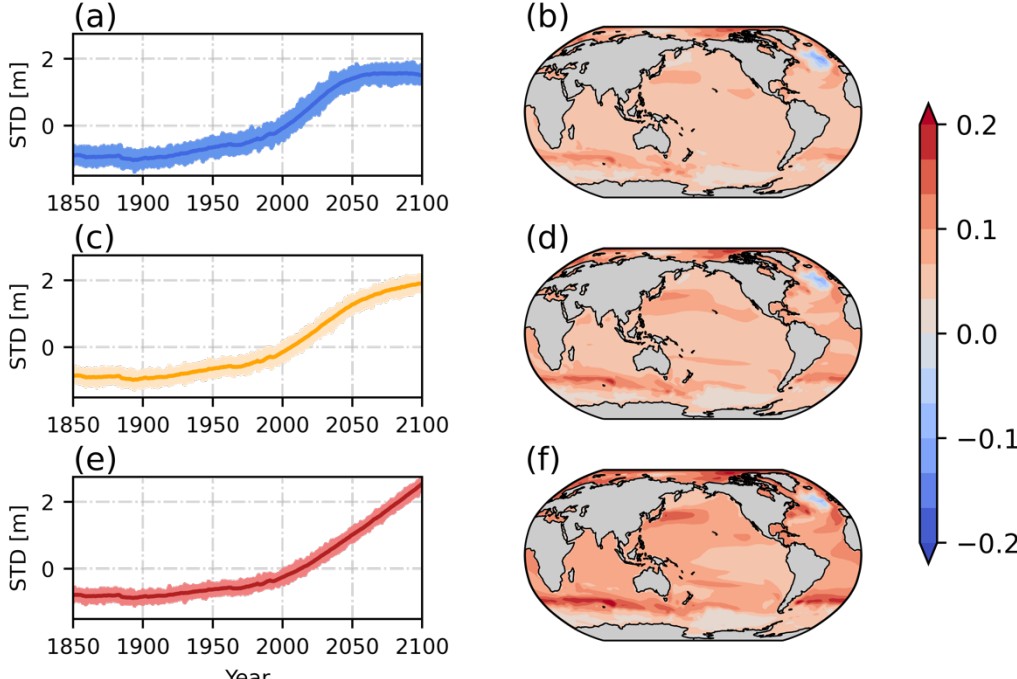

**Figure 2**: Time evolution in terms of SD (a, c, and e, respectively) and associated S/N M EOF pattern number 1 for RCP 2.6, 4.5, and 8.5 (b, d, and f respectively). Light coloured lines in a, c, and d represent SD anomalies from ensemble members, whereas dark coloured lines depict ensemble mean evolution of the pattern. In the historical + RCP scenarios DSL is calculated relative to the mean of 1993–2012

We first compare pattern filtering techniques to conventional methods, in particular an ensemble mean, to isolate the forced response in DSL. For the comparison, we follow the approach used by Wills et al. (2020), where the ensemble is divided into two sub-ensembles: one is used for testing (estimate ensemble) and the other is left for reference (reference ensemble). This leaves us with two 50-member sub-ensembles, where all 50 members in the reference sub-ensemble are used to estimate the forced response by either using ensemble averaging or S/N M EOF pattern filtering and is considered ground truth. The other (estimate) 50-member ensemble is also used to estimate the forced response, but this is performed 49 times by increasing the number of members included in the analysis from 2 to 50. This procedure enables an evaluation of the number of ensemble members needed in the estimate sub-ensemble to characterize the forced response based on explained variance (i.e., r2) in the reference sub-ensemble. To consider sampling uncertainty, this process is repeated ten times for random choices of realizations, taking the median value of all iterations.

When simple averaging is used, we find that 50 members are not sufficient to constrain at least 80% of the forced response variance of the reference ensemble over most of the ocean surface (Fig. 3a). In contrast, S/N M EOF pattern filtering characterises the forced response more efficiently than simply averaging, as it requires a much smaller number of realizations



to remove natural variability (Fig. 3b). While the grid-point median value of the number of ensemble members required is 50
or more when using simple averaging, the median estimate for the filtering method is reduced to eight. Large areas of the
ocean benefit from filtering and there are significant reductions, especially the Indian Ocean, South and Northwest Atlantic
Ocean, as well as large areas in the Pacific Ocean (Fig. 3b). Other areas, however, remain over the 50-member threshold to
explain forced response variance after filtering. Those areas are mostly found where strong western boundary currents exist
(Imawaki et al., 2013), as well as in areas influenced by the Antarctic Circumpolar Current (Rintoul et al., 2001). In those
locations, variability is higher, and a larger number of realizations is needed to characterize it. Yet, there clearly is an advantage
in using S/N M EOF over simple averaging methods, as less realizations are required to explain a significant part of the forced
response in DSL, which means that the forced response can also be determined in models with smaller ensembles.

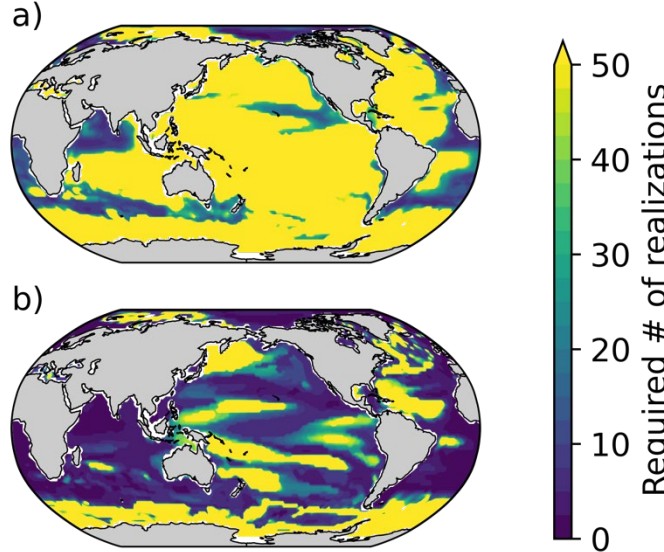


**Figure 3**. The number of ensemble members (realizations) needed to explain at least 80% of the forced response variance
using an ensemble average (a) and using S/N M EOF pattern filtering (b) for RCP 2.6. The reference dataset is an average (a)
or S/M EOF-filtered sub-ensemble (b) of 50 members which does not share realizations with the sub-ensemble used for
estimation. Values represent the median of ten random choices of realizations sampling for both estimate and reference sub-
ensembles. Note that bright yellow indicates more than 50 ensembles members required.
**4.2 Improved Pattern Scaling Using SMILES**
In this section, we demonstrate how S/N M EOF pattern filtering can increase the capabilities of statistical approaches for
explaining DSL based in GMTSLR by reducing internal variability within SMILES. For comparison, we first show pattern
scaling performance when using single realizations and how conventional methods (ensemble mean) reduces RMSE when





using a couple of realizations instead. Second, we examine S/N M EOF as a method for reducing RMSE more efficiently. We
compare regional RSME from both ensemble mean and pattern filtering on only two realizations to allow an assessment of the
areas that benefit the most from filtering when a few simulations are available. Lastly, we contrast how both ensemble mean
and S/N M EOF pattern filtering reduce global mean RMSE as the number of realizations included in the analysis is increased.
As pattern scaling is performed on a grid-point basis, regression performances can be location dependent (Fig. 4a). Despite
such regional variations, we found not substantial differences between GHG scenarios for both the regional and global mean
RMSE estimates when pattern scaling DSL simulations extending up to 2100. Thus, results shown and discussed here are
pertinent to the historical+RCP2.6 scenario for illustrative purposes, unless otherwise stated. When applying pattern scaling
on a single realization of DSL from MPI-GE, the area-weighted, ensemble average RMSE is 3.78 cm, a value which is similar
to previous estimates from studies performed on some of the CMIP5 models (Bilbao et al., 2015; Yuan and Kopp, 2021).
However, pattern scaling performance shows a large spatial variability, ranging from 1.13 to 14.95 cm regionally (Fig. 4a).
High RMSE values (i.e., lower regression performance) can be found in places subject to non-linear mesoscale processes
driven by strong currents, coinciding with the places where the S/N M EOF technique requires many realizations to explain at
least 80% of the forced response variance (Fig. 3b). These are the Antarctic Circumpolar Current (Southern Ocean) or western
boundary currents, including the Gulf Stream (West North Atlantic), and Agulhas Current (South Africa), the Kuroshio Current
(West North Pacific), and at the Brazil-Malvinas Confluence (West South Atlantic). Low RMSE values are found in the more
stable eastern boundary currents, such as the Humboldt (Peru) Current, and in equatorial locations where DSL is relatively less
influenced by large modes of climate variability (e.g., Equatorial Atlantic and Indian Ocean).
Despite its inefficiency, using an ensemble average cancels out some of the natural variability that varies in a different phase
between realizations. When using a 2-member ensemble mean, RMSE reduction is observed both globally and regionally: The
area-weighted average RMSE estimate is reduced from 3.78 to 2.77 cm (27% reduction) when two ensembles are used, with
regional values ranging from 0.87 to 11.00 cm (Fig. 4b). This translates to increased statistical model capabilities within the
entire model domain. While grid-point RMSE reduction ranges from 10 to 30%, the majority of the ocean benefits from a
decrease of more than 25% due to the removal of some of the internal variability (Fig. 4c). Locations experiencing a lower
improvement in regression performance include those that already performed relatively well prior averaging and those with a
high internal variability.



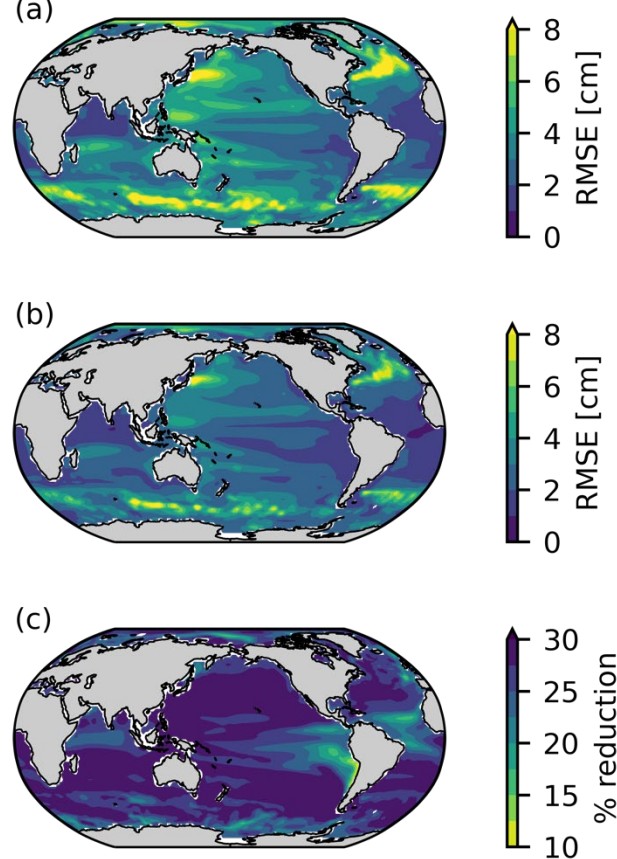


**Figure 4**. Regional pattern scaling performance based on regression RMSE when one realization (a) and a two-member ensemble average (b) are used in the univariate regression. Sampling uncertainty is accounted for in (a) by averaging RMSE from pattern scaling performed individually to the 100 realizations, whereas in (b) random pairs (without replacement) are taken to for the two-member ensemble average. The difference in regression performance between (a) and (b) is shown in (c) in terms of percentage. Results are shown for RCP 2.6 as an example.

To compare how S/N M EOF pattern filtering improves pattern scaling as opposed to averaging, we take two ensemble members from the MPI-GE historical+RCP2.6 experiment and proceed to remove their natural variability by pattern filtering. The 2-member pattern-filtered DSL (Fig. 5a) shows an improved RMSE with similar regional structures compared to its averaged counterpart (Fig. 4b), featuring higher values in western boundary currents and Southern Ocean. Nonetheless, the overall improvement is apparent in all areas: the global estimated RMSE from the regression decreases almost 60% from an average value of 2.77 to 1.12 cm (Fig. 5 c and d). Regionally, RMSE ranges from 0.39 to 6.05 cm when filtering is applied on two ensemble members (Fig. 5a and c). The differences between averaged and filtered approaches are substantial





and location dependent, with filtering yielding a decrease in RMSE ranging from 12% to about 80% (Fig. 5b). The tropical
Indian and Eastern Pacific Ocean are among the locations benefiting the most from the largest performance improvement,
which highlights the skill of pattern filtering to remove variability associated with large climate modes (e.g., ENSO has a
large influence on sea level in the Eastern Pacific Ocean). Similar to previous findings when using averaging (Fig, 4c),
pattern filtering offers a reduced improvement in areas where regression already performed relatively well or where the
presence of meso-scale processes is significant. Regardless of improvement magnitude, pattern filtering provides an overall
increase in regression performance that is observable in the entire ocean domain. While averaging also offers an
enhancement of pattern scaling skill, filtered 2-member pairs produce a distribution of RMSE that is significantly superior
(Fig. 5c).
We further investigate how pattern filtering enhances regression compared to averaging by increasing the number of
members included in the analysis (Fig. 5d). Increasing the number of realizations grants ensemble averaging a considerable
decrease in RSME. Yet, performance improvement asymptotically reaches a plateau around 20 members after which further
reductions in RMSE are modest. Regression based on pattern-filtered DSL also shows an improvement as the number of
realizations increases. Such improvement is very limited compared to the one undergone by averaging, although filtering
always provides a superior performance regardless of the number of members incorporated in the analysis. Importantly, area-
weighted RMSE values differ significantly between the considered approaches when only a small number of realizations are
available and become more similar for a larger number. This highlights the role of pattern filtering techniques when only a
few ensemble members are available. Based on the analysis performed on the DSL simulations from the MPI-GE, filtering
two members provides a regression performance that would only be achieved by averaging at least 12 members.



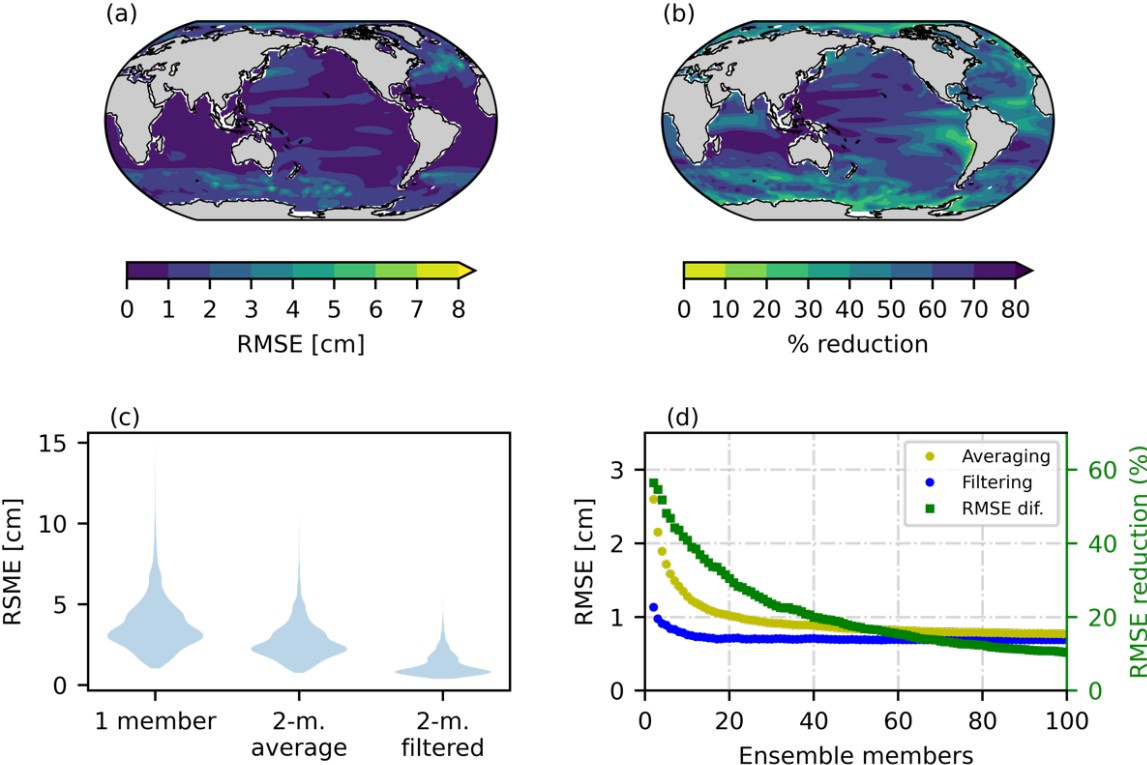

**Figure 5**. Regional pattern scaling performance based on regression RMSE when two ensemble members are used to estimate the forced response via S/N M EOF pattern filtering (a). Panel (b) shows the difference in regression performance between the 2-member average pattern scaling (Fig. 4b) and the S/N M EOF-filtered equivalent (a). Violin plots depicting RMSE distributions from the 1-member, 2-member average, and 2-member S/N M EOF-filtered approaches are shown in panel (c). The area weighted average RMSE obtained in the regression is shown in (d) as a function of the number ensemble members included when using an ensemble mean (yellow) and filtering (blue). The difference in performances in terms of percentage is shown in green. Realizations used here belong to the RCP 2.6 scenario (we observed no discernible differences between scenarios).

## 4.3 Improved Pattern Scaling Using Single Realizations

Most models in CMIP prior to CMIP6 provided only one realization of historical and scenario simulations. Therefore, we now test whether pattern filtering could improve regional emulation of single-realization models. To do so, we apply LFCA which uses a similar approach to S/N M EOF (as explained in Section 3.1.2). In this section, we first examine how LFCA improves the regression RMSE by truncating internal variability in a single simulation from the MPI-GE. We then apply LFCA to a





range of CMIP5 models that were used in previous patterns scaling analyses of DSL, focusing on the differences between
models and RCP scenarios in longer simulations.
LFCA filtering uses the same linear algebra machinery as S/N M EOF, providing a similar regional improvement in pattern
scaling (compare Fig. 5a and 6a). Slightly higher RMSE values are observed in LFCA-based regression, for instance, in the
equatorial Pacific. This is expected because only one simulation is used, compared to two simulations in S/N M EOF filtering,
which enables the latter to identify a larger proportion of internal variability. LFCA provides a substantial reduction in RMSE,
as compared to using a single simulation in pattern scaling (Fig. 6b-c). Regionally, it shows a similar qualitative pattern of
improvement as the other methods shown here (Fig. 6b vs 4c and 5b; averaging and S/N M EOF filtering, respectively).
Quantitatively, however, LFCA provides a larger RMSE reduction on a single realization than S/N M EOF performed on two.
LFCA provides a reduction of the area weighted average RMSE of 68% for all radiative forcing scenarios (Fig. 6c), while S/N
M EOF yields 67% when using two realizations relative to unfiltered 1-member pattern scaling. While both estimates are quite
similar, it is worth noting that S/N M EOF requires two ensemble members to provide such reduction, while LFCA leads to a
similar performance just using one simulation. Similar to S/N M EOF pattern filtering, no substantial differences are found in
pattern scaling RMSE between RCP scenarios up to 2100 (Fig., 6c). This implies that the relationship between DSL and
GMTSLR is analogous between RCP scenarios, hence, a linear regression for projecting DSL leads to a similar performance
for all RCPs both globally (Fig. 6c) and regionally (not shown).



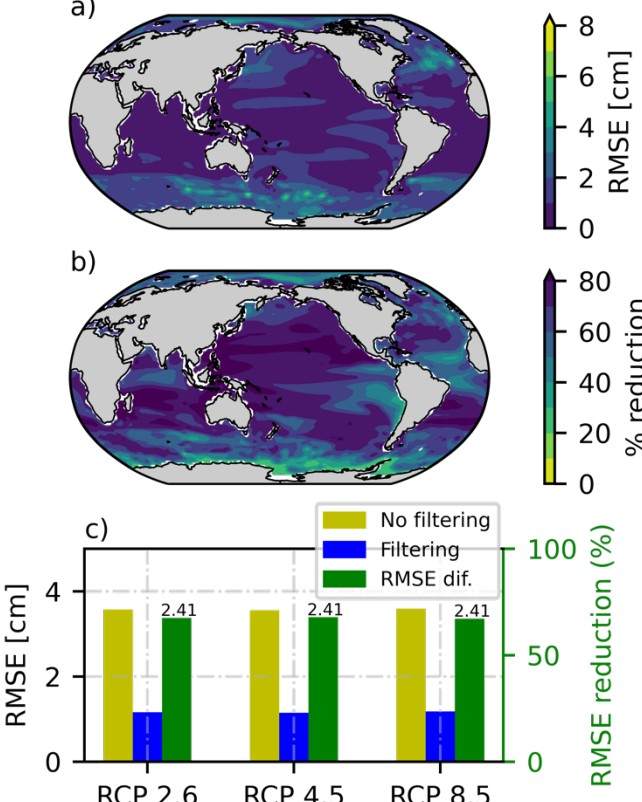

**Figure 6**. Regional pattern scaling performance based on regression RMSE when one (RCP 2.6) ensemble member is filtered via LFCA (a). Filtering is performed individually for each ensemble member to compute 100 scaling patterns whose results are averaged to diminish sampling issues. Differences in regression performance between Fig. 4a (unfiltered 1-member pattern scaling) and (a) are shown in (b) in terms of percentage. The area-weighted average RMSE is shown in (c) for RCPs 2.6, 4.5, and 8.5 and depending on whether the ensemble member is (blue) or not (yellow) filtered. Green indicates RMSE reduction between approaches in terms of percentage, whereas values on top of the bars are the absolute differences in cm.

We further explore the performance of LFCA by comparing the pattern scaling results when isolating the forced response for other GCMs. We identify the forced DSL in four CMIP5 models, being GISS-E2-R, HadGEM2-ES, IPSL-CM5A-LR, and MPI-ESM-LR (Fig. 7a-d, respectively), which all provide scenario simulations up to 2300. To ease comparison with results from the MPI-GE, however, we first examine results up to 2100 (Fig. 7a-d, small r.h.s. insets). RMSE from unfiltered simulations up to 2100 vary between models, and so does RMSE reduction provided by LFCA. Nonetheless, error reduction within a model and between scenarios is very similar, as previously observed for the MPI-GE. This implies that, for all models considered here, there are no significant changing behaviours in the relationship between DSL and GMTLSR between RCP scenarios up to 2100.





When considering results up to 2300, pattern scaling of unfiltered DSL against GMTSLR yields similar results as previous
studies (Bilbao et al., 2015), showing a global area-weighted mean RMSE between 2 and 4 cm. RMSE in both unfiltered and
filtered simulations of DSL increases with radiative forcing for all models considered. As simulations run up to 2300, a
decrease in pattern scaling performance for higher RCPs may indicate a more important role of the deeper ocean layer driving
non-linear processes (Bilbao et al., 2015; Yuan and Kopp, 2021). This tendency is also reflected in the error reduction after
filtering, which decreases as radiative forcing increases both over time and because of the higher emissions scenario, but the
latter is more apparent. Although LCFA filtering improves the performance of pattern scaling for all four CMIP5 models,
considerable differences in error reductions are observed. For instance, HadGEM2-ES benefits the most from pattern filtering
between all the models, with a ~70% decrease in error for RCP 2.6. Conversely, GISS-E2-R undergoes the lowest reduction
after pattern filtering, with about a 50% increase in performance for the same RCP scenario. Differences in model performance
pre- and post-filtering do not only highlight differences in how natural variability is represented in distinct models but may
also reflect model differences in terms of physics representation and modelled forced response.

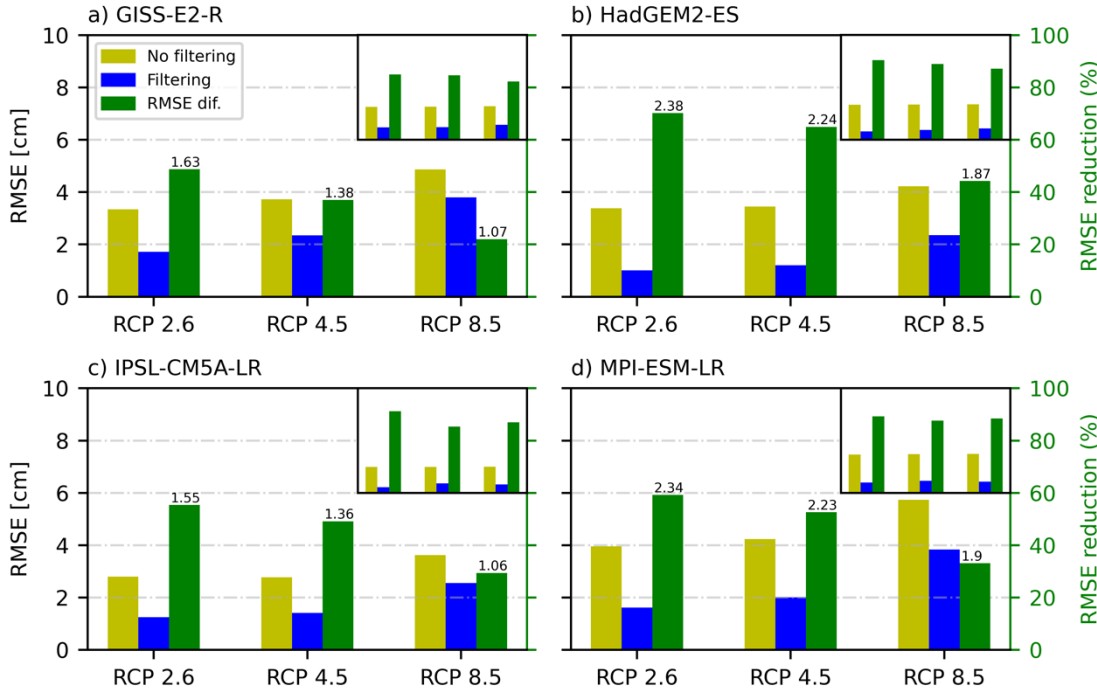


**Figure 7**. Area-weighted average RMSE is shown for RCP 2.6, 4.5, and 8.5 and depending on whether the ensemble member

is (blue) or is not (yellow) filtered via LFCA. Green indicates RMSE reduction between approaches in terms of percentage,
whereas values on top of the bars are the absolute differences in cm. Different panels represent different CMIP5 models





considered here, as stated on top of each panel. The main panel includes simulation data up to 2300, whereas the small inset
on the right-hand top corner shows RMSE results up to 2100.

## 5 Conclusions

Regional emulation tools for DSL change are complementary approaches to GCMs that allow for computationally cheap
statistical projections. Most DSL regional emulators are based on pattern scaling, a statistical model usually based on a grid-
point regression against a global variable representing change in the climate system driven by external forcing. While choosing
suitable global predictors is essential for appropriate tuning of the statistical model, random errors can remain leading to high
uncertainties in statistically based projections. A portion of these random errors are driven by internal variability in DSL and
can be characterised using macro-initialized initial condition large ensembles (SMILES), which are designed to facilitate a
separation between internal variability and external forcings within a model. Here, we applied pattern recognition techniques
to a SMILE with the aim to efficiently truncate internal variability and demonstrate how these approaches could significantly
reduce random errors in regional emulators of DSL.
Although internal variability can be also reduced by using more conventional methods, such as computing an ensemble mean
or linear trends, this requires a relatively large number of realizations to do it effectively. This is a significant constraint
particularly for modelling experiments featuring a limited number of realizations. A more efficient alternative consists of
employing methods that exploit spatial covariance information, such as S/N M EOF pattern filtering. We have demonstrated
that S/N M EOF applied to two realizations attains the same level of error reduction as averaging 12 realizations. The largest
improvement relative to unfiltered simulations was observed when only a few simulations were available, whereas both S/N-
filtered and ensemble average model performance tended to converge for a large number of ensemble members. By identifying
spatiotemporal coherent structures, the S/N M EOF filtering was particularly skilful at removing internal variability due to
large modes of climate variability, such as the ENSO influence on sea level in the Eastern Pacific.
S/N M EOF pattern filtering can identify the common response within at least two realizations. This motivated us to also test
LFCA, which can remove variability in single relalization modelling experiments by applying a lowpass filter. Apart from
being computationally more efficient, LFCA outperforms S/N M EOF in improving the performance of DSL pattern scaling
when using one or two realizations. However, previous studies have emphasized that S/N M EOF pattern filtering provides a
range of benefits compared to LFCA, including: 1) a better isolation of the forced response when the number of ensemble
members is large, and 2) the detection of relatively less important forced patterns, such as those driven by volcanism.
We have also investigated LFCA by applying it to longer (up to 2300) CMIP5 simulations. We found that pattern scaling
performance is independent of the GHG emission scenario up to 2100 and decreases with radiative forcing beyond 2100. Since
we used a linear model, this implies that non-linear processes have different effects on DSL depending on the GHG scenario
and this is reflected in a decrease in model performance depending on the emissions. We also found substantial differences
between CMIP5 models, due to variability being represented differently as well as distinct model physics. Nonetheless, the



performance improvement of pattern scaling when applying LFCA filtering is considerable for all models and scenarios,
ranging from 20% to more than 70% reduction relative to the unfiltered results.
Here, we have demonstrated that reducing internal variability increases the capabilities of statistical approaches to project
DSL. Pattern recognition techniques are especially advantageous for such a task, as they do not require numerous realizations
to significantly reduce uncertainties in statistical projections and no data is lost (as in 30-year means) when reducing internal
variability. Previous studies have not considered removing internal variability prior to searching for suitable global predictors,
which could significantly reduce uncertainties in statistically projected DSL. Hence, for future emulation studies of DSL, we
recommend pattern filtering as a pre-processing step before selecting suitable predictors.

## Code availability

The methods used to perform this study are an adaptation from the ones used by Wills et al. (2020). The code is available at
https://github.com/rcjwills/forced-patterns and https://github.com/rcjwills/lfca.

## Data availability

Simulations from the MPI-GE can be obtained at https://esgf-data.dkrz.de/projects/mpi-ge/, whereas CMIP5 data can be found
at https://esgf-node.llnl.gov/search/cmip5/.

## Author contribution

VMS devised, designed, and performed the analysis, and wrote the manuscript. ABAS supervised the study and contributed
to writing. THJH contributed to data pre-processing and manuscript writing. SD and MM provided valuable feedback on
methods and contributed to writing. NM provided useful information on the use of the MPI-GE.

## Competing interests

The authors declare that they have no conflict of interest.

## Acknowledgements

VMS, ABAS, THJH were supported by PROTECT. This project has received funding from the European Union's Horizon
2020 research and innovation programme under grant agreement No 869304. SD acknowledges David and Jane Flowerree for
their support. We acknowledge the World Climate Research Programme's Working Group on Coupled Modelling, which is
responsible for CMIP, and we thank the climate modelling groups for producing and making available their model output. For



CMIP the U.S. Department of Energy's Program for Climate Model Diagnosis and Intercomparison provides coordinating
support and led development of software infrastructure in partnership with the Global Organization for Earth System Science
Portals.

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
