# Peer review of "Improving Statistical Projections of Ocean Dynamic Sea-level Change"

_EGUsphere, 2022_

## Author Response (AR1)

Review of « Improving Statistical Projections of Ocean Dynamic Sea-level Change Using Pattern Recognition Techniques" by Malagon-Santos et al.

The paper investigates the benefit of using pattern recognition approaches to assess statistical regional sea level projections from coupled climate model outputs. The study shows that using EOF pattern recognition and low-frequency component analysis significantly reduce errors in pattern scaling of regional ocean dynamic sea level change. The authors apply those two methods on the large ensemble MPI-GE simulations. Each member has different initial conditions. Therefore, it is possible to assess the impact of ocean initial conditions on projected dynamic sea level change. The presented results highlight the need to apply such a pattern recognition methods to reduce errors in regional emulation tools of ocean dynamic sea level change especially when a few realizations are available because of the huge computation cost.

The topic of the paper is interesting as the future generation of AOGCM will increase both atmosphere and oceans spatial resolutions. Thus, a few simulation integrations will be preferred from large ensembles because of the computational cost. Therefore, this technique may be relevant for future sea level change investigations.

Thank you for the positive comments.

I find the paper well written. It is well organized. However, the methodology part could be improved as the methodology is not easy to understand especially for a non-expert in pattern filtering. I think the authors can provide more explanations to help the reader.

We agree the methodology section may be difficult to follow for a non-expert. We thought it was not needed to further expand the methodological steps, as interested readers are referred to appropriate literature where methods are thoroughly explained already. We have summarized the methodology to include only the basics of both methods. Readers are referred to appropriate literature if they wish to read a more detailed explanation. We've also added a flowchart (now Fig. 1 in the revised manuscript) showcasing how the forced response is constructed and separated from internal climate variability. We hope these changes make the paper seem less technical and more accessible to a wider audience.

See lines 220-271 for the revised Methods.

Overall, the paper is well supported but some parts are unclear. For instance, I struggle to fully understand and interpret Fig1 as it lacks of explanation in the caption (see my comment below).

I find the paper very technical and I wander if Ocean Science is the right journal to publish this piece of work. I recommend a major revision for the manuscript before a possible publication.

Ocean Science was chosen based on the specific dedication of the journal, which is on all aspects of ocean science including experimental, theoretical, and laboratory studies. We believe the topic of manuscript falls within different fields covered by Ocean Sciences, especially ocean physics and ocean models. Moreover, since our paper aims to simplify complex global climate (or related) models by using statistical approximations, the methods and results presented here may be also tested in other oceanic processes, such as regional changes in temperature. This made us feel Ocean Science was a suitable journal for this manuscript. We hope the amended methods sections will make the paper seem less technical and more accessible to a broader audience in Ocean Science.

Major comments

- When using EOF decomposition, one strong assumption is that all the modes are independent (i.e., they are orthogonal to each other). Is it really the case especially at global scale? This might be discussed in the conclusion as a limitation of the approach.

  By construction, the EOFs patterns and principal components are orthogonal. It is this orthogonality constraint what inhibits a physical interpretation of EOFs (as noted in Line 308 of the original manuscript, 293 in the revised manuscript).

- What do you mean by 'well separated'? (L143) How is it performed? Are you sure the initial conditions are totally different and independent? Please, clarify.

  Single-model initial condition large ensembles (SMILES) are designed to assess a range of outcomes due to the presence of unpredictable internal climate variability. This is usually achieved by running a number of simulations with the same model and identical forcing, only differing in their initial conditions. Simulations are independent as long as the memory of the initial conditions is lost, which ensures each ensemble member have a unique climate trajectory (Deser, 2020). There are two main procedures to achieve this: 1) by inducing small round-off level differences in their atmospheric initial conditions (micro-initialization); 2) by branching simulations at different times in the control simulation (macro-initialization). Both micro and macro initialization procedures are useful to characterize unpredictable internal variability within a model. Macro-initialization, however, provides larger differences in the initial states in both the atmosphere and ocean. Since we are assessing ocean processes

here (i.e., ocean memory is important for our analysis), we deemed a macro-initialized ensemble more suitable for the purpose of this study.

MPI-GE simulations assume a stationary and volcano free 1850 climate, and are macro-initialized on the first of January in different years of the control simulation (Table 1 in Maher et al., 2019). The branching separation between realizations varies along the pre-industrial control, ranging from 6 to 24 years and with a median of 16 years.

As already noted in Line 144 (original manuscript), the branching times of macro ensembles are designed to sample large scale aspects of the climate systems (atmosphere, land, and ocean). Nonetheless, we have included further information on the MPI-GE in the revised manuscript and emphasize its value for assessing internal climate variability within the model. We have also provided an improved description of micro vs macro and why we chose the latter. (See lines 145-161 for these amendments)

Deser, C. (2020). Certain uncertainty: The role of internal climate variability in projections of regional climate change and risk management. Earth's Future, 8(12), e2020EF001854.

Maher, N., Milinski, S., Suarez-Gutierrez, L., Botzet, M., Dobrynin, M., Kornblueh, L., ... & Marotzke, J. (2019). The Max Planck Institute Grand Ensemble: enabling the exploration of climate system variability. Journal of Advances in Modeling Earth Systems, 11(7), 2050-2069.

- As GMTSLR is removed, the underline hypothesis is that the model conserves volume instead of mass. Is that right? If so, this is due to the Boussinesq's approximation. This should be clearly stated to avoid any misunderstanding.

We would like to clarify GMTSLR is not removed, we simply do not use it. Dynamic sea level (zos) and GMTSLR (zostoga) are usually provided separately in AOGCMs.

Almost all CMIP6 and CMIP5 models use the Boussinesq approximation (Irving et al., 2021) which implies, as mentioned by the reviewer, that volume is conserved instead of mass. This means steric processes are represented by a change in density, from which a change in mass can be inferred (the so-called Boussinesq ocean mass). That is why GMTSLR (or zostoga) is inferred separately.

Irving, D., Hobbs, W., Church, J., & Zika, J. (2021). A mass and energy conservation analysis of drift in the CMIP6 ensemble. Journal of Climate, 34(8), 3157-3170.

• MPI-GE description is too succinct. Please, provide more insights. There is no mention on the spatial resolution of the MPI-GE simulations especially for the ocean part. I assume that the ocean spatial resolution is about 1° meaning that the oceans have laminar flows. If so, what is the consequence when assessing the internal variability? Are not you underestimated it? Some studies have estimated the ocean-based internal variability from a large ensemble of forced OGCM. When increasing the spatial ocean resolution, the ocean-based internal variability increases in space and time. We can expect the same behavior for the coupled internal variability. I would appreciate some discussion on this specific point in the discussion's section.

The model is indeed course resolution: T63L47/GR15L40 ("LR" - Low Resolution). Nonetheless, Suarez-Gutierrez et al. (2021) show that MPI-GE well samples observed ocean variability in all regions except for the Southern Ocean. Below we provide more details about the model's resolution:

• Atmosphere: approximate horizontal resolution of 200 km (1.875 degrees) at 47 layers (up to 0.01 hPa / 80 km in height)
• Land biosphere (interactive vegetation): same horizontal resolution as atmosphere.
• Ocean including biogeochemistry: horizontal resolution varies from 12 to 150 km at 40 layers.

This information is now in lines 145-161 in the revised manuscript.

It is true that variability tends to become larger at higher model resolutions (Penduff, 2010). However, since the goal of our study is to remove internal variability and isolate the forced response to improve its statistical modelling, how well internal variability is represented in a model should not be important in our analysis.

Penduff, T., Juza, M., Brodeau, L., Smith, G. C., Barnier, B., Molines, J. M., ... & Madec, G. (2010). Impact of global ocean model resolution on sea-level variability with emphasis on interannual time scales. Ocean Science, 6(1), 269-284.

Suarez-Gutierrez, L., Milinski, S., & Maher, N. (2021). Exploiting large ensembles for a better yet simpler climate model evaluation. Climate Dynamics, 57(9-10), 2557-2580.

Minor comments

L54-63: When describing the drivers of regional sea level changes, one might want to know the associated time scales of each processes. Please, clarify. This would help the reader.

We agree with the reviewer that this could help the reader notice the differences between the different regional contributions and highlight the importance of ocean dynamics as a significant driver of variability in sea-level change projections. Some examples and appropriate literature have been added in line 63 in the revised manuscript.

Durand, G., van den Broeke, M. R., Le Cozannet, G., Edwards, T. L., Holland, P. R., Jourdain, N. C., ... & Chapuis, A. (2022). Sea-level rise: From global perspectives to local services. Frontiers in marine science, 8, 2088.

L66: What do you mean by natural variability? Could you define this concept? This would help the readers.

As also pointed out by the other reviewer, we use 'natural variability' here when we meant to refer to 'internal climate variability'. We realized we committed a mistake when using those terms interchangeable throughout the paper, which can lead to confusion. Climate variability is defined as variations in the mean state and other statistics (e.g., extremes) of the climate (Mason-Delmotte et al., 2018). Climate variability can be caused by natural internal processes (internal variability) or by variations in natural or anthropogenic external forcing (external variability). In this study, we address internal climate variability, defined as naturally occurring climatic variations controlled by interactions between different components of the Earth system (Hasselmann, 1976; Schwarzwald et al., 2022). We have checked the paper and made terminology consistent to avoid ambiguity, including a definition of internal climate variability as suggested (see line 71)

IPCC, 2018: Annex I: Glossary [Matthews, J.B.R. (ed.)]. In: *Global Warming of 1.5°C. An IPCC Special Report on the impacts of global warming of 1.5°C above pre-industrial levels and related global greenhouse gas emission pathways, in the context of strengthening the global response to the threat of climate change, sustainable development, and efforts to eradicate poverty* [Masson-Delmotte, V., P. Zhai, H.-O. Pörtner, D. Roberts, J. Skea, P.R. Shukla, A. Pirani, W. Moufouma-Okia, C. Péan, R. Pidcock, S. Connors, J.B.R. Matthews, Y. Chen, X. Zhou, M.I. Gomis, E. Lonnoy, T. Maycock, M. Tignor, and T. Waterfield (eds.)]. Cambridge University Press, Cambridge, UK and New York, NY, USA, pp. 541-562, doi:10.1017/9781009157940.008.

Hasselmann, K. (1976). Stochastic climate models part I. Theory. tellus, 28(6), 473-485.

Schwarzwald, Kevin, and Nathan Lenssen. "The importance of internal climate variability in climate impact projections." Proceedings of the National Academy of Sciences 119.42 (2022): e2208095119.

L75: What do you mean by 'regional emulation tools'? Please, define any new terminology.

Emulation is a method consisting of parameterizing process-based models so that their output is estimated at significantly reduced computational cost (Thomas and Lin, 2018). Regional emulation follows the same principle and aims to estimate a spatiotemporal varying variable by mimicking computationally expensive approaches, such as process based GCMs, using less computationally extensive approaches, such as statistical models. We have included this definition here and provide appropriate references and examples when doing so (see line 80).

Thomas, M. A., & Lin, T. (2018). A dual model for emulation of thermosteric and dynamic sea-level change. Climatic Change, 148(1-2), 311-324.

L109-110: How many members do you need to completely cancel out the internal variability?

There is no straightforward answer for this question. Strictly speaking, we would need an infinite number of realizations to completely cancel out variability. The number of members needed to robustly characterize internal variability depends on the question to address and acceptable error, as explained by Milinski et al. (2020). What we have observed in this study regarding dynamic sea-level variability is that internal variability associated to well-known modes of climatic oscillations leading to coherent spatial structures may be easy to define using a few ensemble members, whereas higher variability (e.g., related to eddy dynamics) is much more difficult identify.

Milinski et al 2020: How large does a large ensemble need to me. https://doi.org/10.5194/esd-11-885-2020

L297: What do you mean by 'conventional approaches'? please, clarify.

We refer to simpler but less efficient approaches that have been widely used to remove internal variability, such as ensemble averaging. This has been referred to in other sentences, (e.g., 27, 110, 333, 365, 502) but will also include it here to increase readability.

L323-324: '…that appear to be linked to volcanic eruptions'. Can you bring extra explanation here or a suitable reference?

The time evolution (in standard deviation) of these patterns is rather stable except for specific points in time where aerosol forcing was significantly altered in the atmosphere due to volcanic eruptions. Aerosols from volcano eruptions can change temperatures in the atmosphere (Wills et al., 2020), which in turn also affects sea level. As an example, the peaks that can be seen in the figure below (a, c, e) coincide with eruptions from Krakatoa, Agung, El Chinchón, and Pinatubo, suggesting that those eruptions indeed exerted a change in ocean dynamic sea level. We have included this explanation in this in text (line 307)

[Figure]

Figure1: I do not fully understand this plot. Why Sk is decreasing when pattern number is increasing? Please, clarify it and maybe extend the caption.

In the pattern filtering approaches tested here, signal to noise (S/N) patterns are sorted in terms of their signal fractions (Sk), and that is why Sk is maximal for pattern 1 and decreases with pattern number. This is already briefly mentioned in line 233 (in the original manuscript). The caption of this figure now emphasizes this in the amended manuscript. This is also illustrated in the newly introduced flowchart in Figure 1 of the revised manuscript.

Figure2: Please, change SD by standard deviation. This would help the reader.

SD have been changed in the revised manuscript to standard deviation.

Figure 4: Are the results consistent when considering RCP 4.5 an RCP 8.5? It would be interesting to add them into the supplementary materials.

As noted in Sentence 371, we found no significant differences between scenarios when it comes to RMSE reduction provided by pattern filtering techniques. As filtering techniques remove internal variability, and the latter does not significantly change between scenarios up to 2100, results are similar for different scenarios. However, we did found contrast in emulated DSL and the slope difference between unfiltered and filtered results for different RCP scenarios. Emulated DSL differences increases with forcing, and this is expected because so does the magnitude of the predictor (GMTSLR). On the other hand, slope differences are highest for the lowest emission scenarios, and decrease as radiative forcing increasing. Since lower radiative forcing leads to a lower signal/noise ratio, noise (internal variability) can drive large differences in slopes between filtered and unfiltered results (and vice versa).

We have included a new plot (Figure 8 in the revised manuscript) showing differences in emulated DSL for distinct RCPs considered here. We have also added slope differences in supplementary material, together with a few local examples (Fig. S8 to S14 in the revised supplementary material). A discussion of these results can be found in lines 489 -508 of the revised manuscript. We have also added this in the conclusions.

In this study the authors used large ensemble simulations from one climate model to test to what extent pattern filtering approaches help to reduce internal variability in the dynamic sea level. They then discussed the benefits of using such approach to reduce uncertainties in pattern scaling of dynamic sea level change. This is an important research topic as large ensemble simulations are computationally expensive and usually we need to deal with limited or even single ensemble from climate model.

My main comment is that the reduced regression errors (residuals) in pattern scaling after applying the pattern filtering approach are well expected as the internal variability is reduced. I agree quantifying them is useful but the current manuscript fails to demonstrate more value for using such approach prior to pattern scaling, as claimed in the title and main message. Specifically, to what extent the application of pattern filtering could change the slope α of pattern scaling? Is there a significant change? Could you please show this change not only for global maps but also for time series in key regions as examples? Afterall this is what we really obtain and need from pattern scaling.

We thank the referee for their constructive comments. We agree that the comparison between the slopes α from raw and filtered simulations could further highlight the benefit of using pattern filtering approaches. We see substantial slope differences in places subject to non-linear mesoscale processes, such as strong western boundary currents (Fig. 1 in this document) (e.g., Gulf Stream and Kuroshio current; US east coast and Japan east coast, respectively). The maximum slope difference decreases with radiative forcing. Since lower radiative forcing means lower signal/noise ratio, noise (internal variability) can drive large differences in slopes between filtered and unfiltered results. On the contrary, a higher emission scenario is characterized by a higher signal/noise ratio, as noise exerts a less important control on slope differences.

[Figure]

Figure 1. Maximum slope difference between unfiltered and LFCA-filtered realizations, considering all 100 MPI-GE members, for RCP26, RCP45, and RCP85 (a, b, and c, respectively). Black dot to the east of Japan represents location for figures 2, 3, and 4.

The interpretation of slope differences is not straightforward though, so we opted to include and assessment of the differences in emulated DSL in 2100 between unfiltered and filtered data, as this also considers possible differences in the intercept (see Figure 8 and lines 448 – 474 in revised manuscript.). Nonetheless, we have also included a small discussion (Lines 475 – 484) about the difference in slope and a supplementary figure (Figure S8), as well as mentioning this related results in the conclusions (line 538). Also, as suggested by the reviewer, we have added in supplementary material (see Figures S9 to S14 in revised manuscript) the linear fit for key regions where changes in slope are substantial. As an example, we are including here a point in the Kuroshio current (east coast of Japan, see black dot in Fig. 1 in this document)., comparing an unfiltered, a 30-yr moving mean, and a filtered realization for RCP 2.6, 4.5 and 8.5 (Figs. 2, 3, and 4 in this document; respectively). Fig. 2 shows that while unfiltered and 30-year moving means are quite similar, the filtered case shows a positive and much steeper slope (for a single realization, as an example). These differences are caused by internally generated variability and the removal of data points to compute the 30-year mean to remove part of the temporal variability (this is highlighted now in line 479 of the revised

manuscript). Slope differences get smaller as radiative forcing increases (Fig. 2 vs 3 vs 4), as also shown in Fig. 1

[Figure]

Fig. 2. Linear regression model of dynamic sea level (DSL) and GMTSLR for an unfiltered, a 30-year moving mean, and an LFCA-filtered RCP 2.6 realization (left, middle, and right panel, respectively).

[Figure]

Fig. 3. Linear regression model of dynamic sea level and GMTSLR for an unfiltered, a 30-year moving mean, and an LFCA-filtered RCP 4.5 realization (left, middle, and right panel, respectively).

[Figure]

Fig. 4. Linear regression model of dynamic sea level and GMTSLR for an unfiltered, a 30-year moving mean, and an LFCA-filtered RCP 8.5 realization (left, middle, and right panel, respectively).

Some minor comments below,

L21 "model disagreement" is not straightforward here – please consider rephrasing. In the context of last sentence does it refer to "climate model" or "statistical model"? should "disagreement" be "uncertainty" here?

We refer to disagreement between statistically modelled ocean dynamic sea-level change and simulations coming from the respective GMC. To avoid confusion, 'model disagreement' has been changed to 'statistical model error'.

L26 "MPI-GE" might not be familiar to some readers

MPI-GE is now introduced as "Max Planck Institute Grand Ensemble (MPI-GE)" both in the abstract and main text in the revised manuscript.

L26 "so that internal variability is optimally characterized while avoiding model biases" – please consider rephrasing. We can never avoid the model bias issue. My understanding is when using single model large ensemble simulations, the externally forced signal is optimally characterized, which provides important basis to test pattern filtering methods.

The reviewer is right about model biases: it will still be a problem even when using a single model. What we were trying to emphasize here is the benefit of using single-model large ensembles instead of utilizing same-forcing simulations from different models. The former allows us to optimally characterize the externally forced response within a model, whereas the latter could include model biases as externally forced response. We have rephrased this sentence to emphasize large ensemble simulations allows to optimally characterize the externally forced signal within a model and forcing scenario, instead of saying that using them allows us to avoid model biases.

L27 "pattern filtering" do you mean the "two pattern recognition methods (L23)" or specifically the "signal-to-noise maximizing EOF pattern filtering (L24)".

We refer to both methods. We have clarified this in the text.

L66 "natural" should be "internal climate" as used in most other places – please check throughout the manuscript for this.

We agree with the reviewer that, as written in the original paper, natural and internal climate variability seem interchangeable when they are not. We have checked these terms throughout the paper and made modifications accordingly. We have also included a definition of internal climate variability (see line 72) as suggested by the other reviewer, reducing ambiguity. In addition, we noticed we refer to internal climate variability many times, so no we introduce the abbreviation ICV to increase readability.

Figure 3 It's unclear (1) how the number of ensembles needed is calculated; (2) what does "forced response variance" refer to. Could you please make connections to equations in section 3?

First, we would like to clarify that we calculated the required number of ensemble members (realizations), a not the number of ensembles needed. (1) The number of ensemble members needed to explain a certain level of variance of the forced response is based on the coefficient of determination r2 between the two datasets considered. Here, chose the 80% of the variance following similar studies, but other arbitrary level could be chosen. The procedure we to took is as follows:

i.    Create two subsets of the 100-member ensemble, with 50 members each.
ii.   The forced response is estimate from one of the 50-member ensembles using all members in the subset. We used this forced response as reference.
iii.  The forced response is also calculated from the other 50-member subset but instead using all 50 members as in (ii) the number of members is increased from 2 to 50 in an iterative process. We call this subset the testing subset, as it is the one used to estimate the number of ensembles needed to explain a certain level of variance in the reference (step ii) subset.
iv.   The number of required members is computed as follows. We start with only 2 members, which are used to estimate the forced response in the testing subset. We compare both forced responses (2-member testing subset vs 50-member reference subset) by means of the coefficient of determination (r2) which tells us about the proportion of variance that is shared between the two subsets. We do this on a grid-point basis and see where the 80% level is exceeded. For those grid points where the threshold is not exceeded, we do the same comparison but adding an additional member to the testing subset (i.e., 3 members). Again, we check where the 80% threshold is exceeded when an extra member is considered. We continue this procedure by adding more members until we reach 50 members (the maximum in the testing dataset).
v.    We do this comparison when either the forced response is calculated by averaging (Fig. 3a in manuscript) or S/N M EOF pattern filtering (Fig. 3b in manuscript).
vi.   To avoid sampling bias, we repeat this analysis several times by randomizing the initial 100-member ensemble.

We hope it is clearer now. A couple of final notes for this answer.

-    When we say forced response variance, we refer to the proportion of the variance that is shared between the two subsets being compared here. We have clarified this in the text (see line 322 in revised manuscript)

- It is difficult to make connections with section 3 (Methodology), since none of the equations used there has relation to the calculations performed here to estimate the require number of ensembles members to explain the forced response variance within a subset.

Although we attempted to explain how the calculation was performed in the original manuscript, we have expanded such explanation by including some steps of the iterative process (see lines 319 – 336 in the revised manuscript ). We hope this makes the interpretation of the results in Figure 3 (now figure 4) simpler.

---

## Referee Report (RR1)

Review of « Improving Statistical Projections of Ocean Dynamic Sea-level Change Using Pattern Recognition Techniques" by Malagon-Santos et al.

The authors answered all my previous comments and questions. I really appreciate the introduction of Fig1 which help me understand the method. I find the paper improved overall.

I would like to share with the authors only one minor comment I have.

After their answer, I can recommend the paper for possible publication in Ocean Science.

Minor comment

L169-176 : I do not understand the paragraph.

The authors state that: 'zos is defined ... as the difference between local sea-surface height relative to the geoid, and its global mean over the ocean
area (GMTSLR, or '*zostoga*' in CMIP experiments)".

As I understand, the authors consider the global mean thermosteric sea level added to zos at that point.

Then, the authors state: "Hence, by definition, DSL, or *zos*, varies locally due to ocean circulation and horizontal gradients, but its global mean is zero at every time step".

I disagree as the global mean sea level contains the thermosteric contribution which is not equal to zero at each time step.

Do I miss something here? I encourage the authors to clarify the paragraph as it can be misleading.

---

## Author Response (AR2)

Review of « Improving Statistical Projections of Ocean Dynamic Sea-level Change Using Pattern Recognition Techniques" by Malagon-Santos et al.

The authors answered all my previous comments and questions. I really appreciate the introduction of Fig1 which help me understand the method. I find the paper improved overall. I would like to share with the authors only one minor comment I have.

We would like to thank the Referee for their positive feedback and welcome all comments aiming to improve our manuscript.

After their answer, I can recommend the paper for possible publication in Ocean Science. Minor comment

L169-176 : I do not understand the paragraph.

The authors state that: 'zos is defined … as the difference between local sea-surface height relative to the geoid, and its global mean over the ocean area (GMTSLR, or 'zostoga' in CMIP experiments)".

As I understand, the authors consider the global mean thermosteric sea level added to zos at that point.

The purpose of this sentence was to highlight the two main variables that we are using un our analysis, DSL (or zos) and GMTSLR (or zostoga), and not indicate that we were somehow combining them. We agree with the Referee that, in its current form, this sentence can be misleading.

Then, the authors state: "Hence, by definition, DSL, or zos, varies locally due to ocean circulation and horizontal gradients, but its global mean is zero at every time step".

I disagree as the global mean sea level contains the thermosteric contribution which is not equal to zero at each time step.

Do I miss something here? I encourage the authors to clarify the paragraph as it can be misleading.

We have now clarified this paragraph by improving the definition of DSL (zos) and not mentioning GMTSLR (zostoga) to avoid confusion. GMTSRL is now introduced in the following paragraph, where we indicate this variable has been successfully used in previous pattern scaling studies of DSL and express our intention of using it in our study as well (see lines 169 – 183 of the revised manuscript).